# Molecular interaction and inhibition of SARS-CoV-2 binding to the ACE2 receptor

Jinsung Yang[1,4], Simon J. L. Petitjean[1,4], Melanie Koehler [1], Qingrong Zhang[1], Andra C. Dumitru[1], Wenzhang Chen[2], Sylvie Derclaye[1], Stéphane P. Vincent [2], Patrice Soumillion [1] & David Alsteens [1,3 ✉]

Study of the interactions established between the viral glycoproteins and their host receptors is of critical importance for a better understanding of virus entry into cells. The novel coronavirus SARS-CoV-2 entry into host cells is mediated by its spike glycoprotein (S-glycoprotein), and the angiotensin-converting enzyme 2 (ACE2) has been identified as a cellular receptor. Here, we use atomic force microscopy to investigate the mechanisms by which the S-glycoprotein binds to the ACE2 receptor. We demonstrate, both on model surfaces and on living cells, that the receptor binding domain (RBD) serves as the binding interface within the S-glycoprotein with the ACE2 receptor and extract the kinetic and thermodynamic properties of this binding pocket. Altogether, these results provide a picture of the established interaction on living cells. Finally, we test several binding inhibitor peptides targeting the virus early attachment stages, offering new perspectives in the treatment of the SARS-CoV-2 infection.

[1] Louvain Institute of Biomolecular Science and Technology, Université Catholique de Louvain, Louvain-la-Neuve, Belgium. [2] Département de Chimie, Laboratoire de Chimie Bio-Organique, University of Namur, Namur, Belgium. [3] Walloon Excellence in Life sciences and Biotechnology (WELBIO), 1300 Wavre, Belgium. [4] These authors contributed equally: Jinsung Yang, Simon J. L. Petitjean. ✉email: david.alsteens@uclouvain.be

In December 2019, a novel coronavirus (CoV) was determined to be responsible for an outbreak of potentially fatal atypical pneumonia, ultimately defined as coronavirus disease-19 (COVID-19), in Wuhan, China. This novel CoV, termed severe acute respiratory syndrome (SARS)-CoV-2, was found to share similarities with the SARS-CoV that was responsible for the SARS pandemic that occurred in 2002. The resulting outbreak of COVID-19 has emerged as a severe pandemic. The genome of SARS-CoV-2 shares about 80% identity with that of SARS-CoV and is about 96% identical to the bat coronavirus BatCoV RaTG13 (ref. [1]).

CoV entry into host cells is mediated by its transmembrane spike (S) glycoprotein that forms homotrimers protruding from the viral surface[2] (Fig. 1a). The S glycoprotein comprises two functional subunits responsible either for binding to the host cell receptor (S1 subunit including the receptor-binding domain (RBD)) or for fusion of the viral and cellular membranes (S2 subunit). Recent studies claimed that the angiotensin-converting enzyme 2 (ACE2), previously identified as the cellular receptor for SARS-CoV, also acts as a receptor of the new coronavirus (SARS-CoV-2)[3] (Fig. 1b). In the case of SARS-CoV, the S glycoprotein on the virion surface mediates receptor recognition (Fig. 1c) and membrane fusion[4,5]. Recently, the high-resolution cryo-electron microscopy structure obtained on the full-length human ACE2 in the presence of the RBD of the S glycoprotein of SARS-CoV-2 suggests simultaneous binding of two S-glycoprotein trimers to an ACE2 dimer[3]. The S2 subunit is further cleaved by host proteases located immediately upstream of the fusion peptide[6], leading to the activation of the glycoprotein that undergoes extensive irreversible conformational changes facilitating the membrane fusion process. Altogether, the information obtained so far highlights the fact that CoV entry into susceptible cells is a complex process that requires the concerted action of receptor binding and proteolytic activation of the S glycoprotein at the host cell surface to finally promote virus–cell membrane fusion. However, so far, direct evidence about the dynamics of the binding of the S1- to the ACE2 receptor at the single-molecule level is missing.

Here, we analyze the biophysical properties of the SARS-CoV-2 S-glycoprotein binding, on model surfaces and on living cells, to ACE2 receptors using force–distance (FD) curve-based atomic force microscopy (FD-curve-based AFM) (Fig. 1c). We extract the kinetics and thermodynamics of the interactions established in vitro, and compare the binding properties of both the S1 subunit and RBD. Next, we test short ACE2-derived peptides targeting the viral S glycoprotein as potent binding inhibitor peptides and observe a significant reduction in the binding properties.

## Results

**S1 subunit specifically binds to purified ACE2 receptors.** As SARS-CoV-2 binding to ACE2 receptors is thought to play a key role in the first binding step at the cellular membrane[3], we first

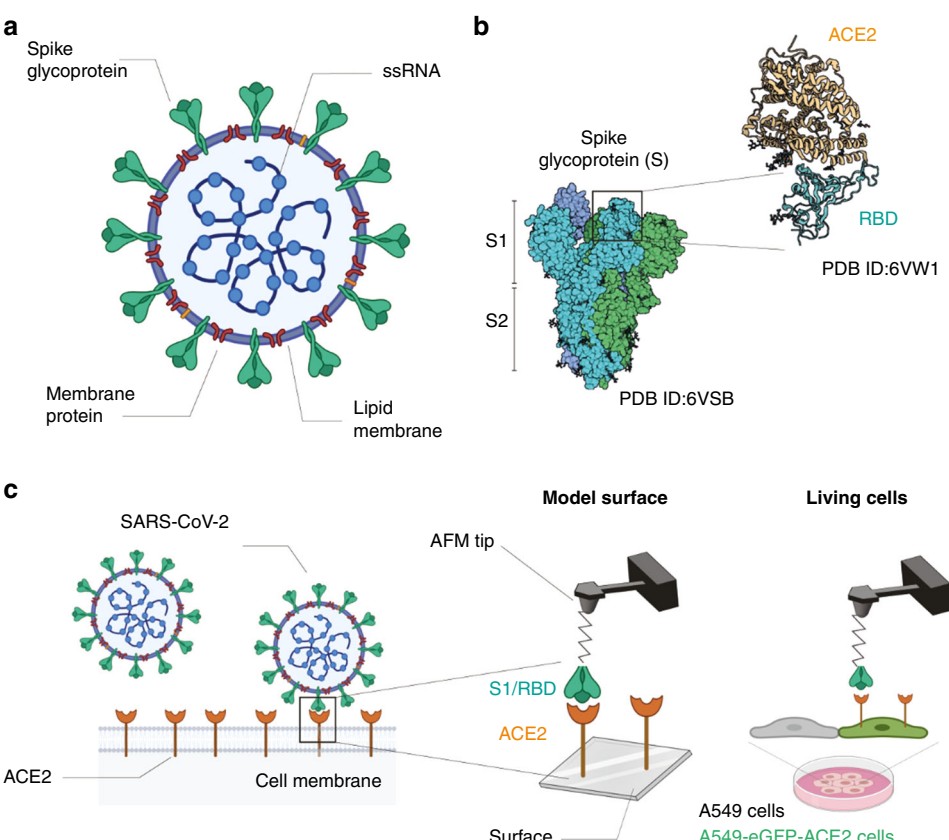

**Fig. 1 Probing SARS-CoV-2 binding to the ACE2 host receptor. a** Schematic of a SARS-CoV-2 particle, an enveloped ssRNA virus expressing at its surface the spike glycoprotein (S) that mediates the binding to host cells. **b** Structural studies have previously obtained a complex between the receptor-binding domain (RBD, a subunit of the S glycoprotein) and the angiotensin-converting enzyme 2 (ACE2) receptor. **c** Schematic of probing SARS-CoV-2 binding using atomic force microscopy (AFM). The initial attachment of SARS-CoV-2 to cells involves specific binding between the viral S glycoprotein and the cellular receptor, ACE2. The interactions are monitored by AFM on model surfaces, where the ACE2 receptor is attached to a surface and the S1 subunit or the RBD onto the AFM tip, and on A549 living cells expressing or not fluorescently labeled ACE2.

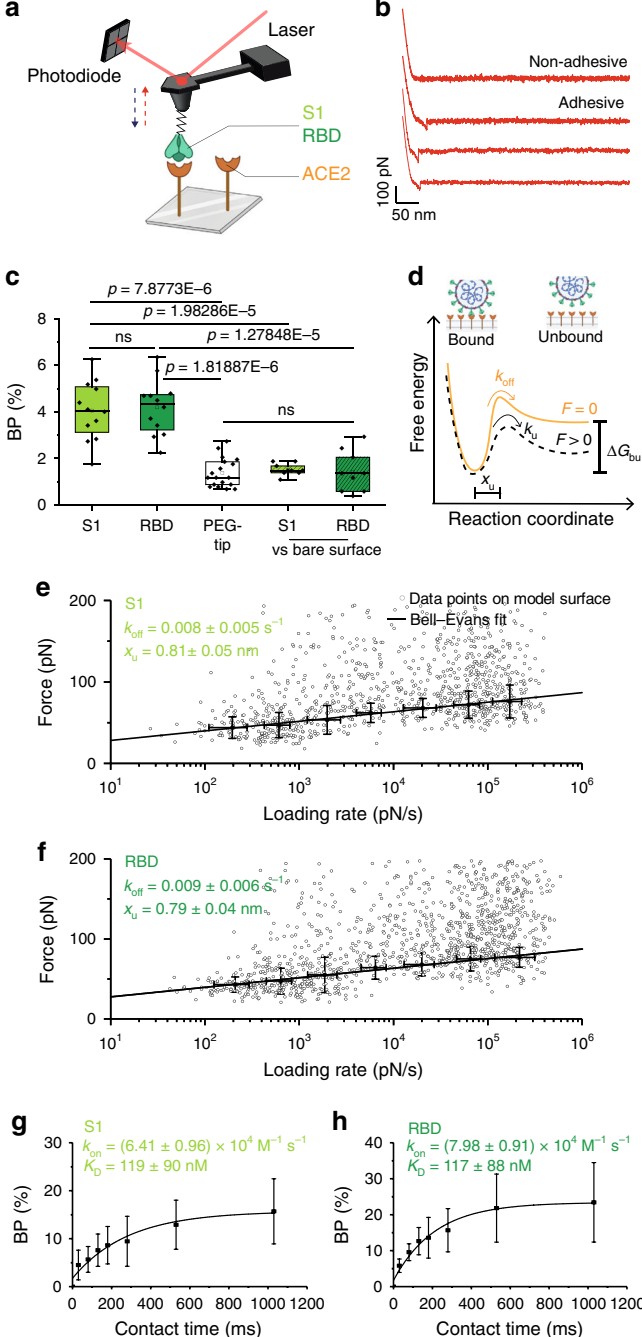

**Fig. 2 Probing S-glycoprotein binding to the ACE2 host receptor on model surface. a** Binding of S-glycoprotein subunit (S1 or RBD) is probed on an ACE2-coated surface. **b** Retraction part of four force–distance curves showing either nonadhesive or specific adhesive curves. **c** Box plot of specific binding probabilities (BP) measured by AFM between the functionalized tip (S1, RBD, or PEG) and the grafted surface (ACE2 or OH-/COOH-terminated alkanethiol (bare surface)). One data point belongs to the BP from one map acquired at 1 μm/s retraction speed. The square in the box indicates mean, the colored box indicates the 25th and 75th percentiles, and the whiskers indicate the highest and the lowest values of the results. The line in the box indicates median. $N = 12$ (S1, RBD), 18 (PEG), and 9 (S1, RBD vs. bare surface) maps examined over 4 (S1, RBD), 6 (PEG), and 3 (S1, RBD vs. bare surface) independent experiments. **d** Bell–Evans model describing a virus-receptor bond as a two-state model. The bound state is separated from the unbound state by a single energy barrier located at distance $x_u$. $k_{off}$ and $k_{on}$ represent the dissociation and association rate, respectively. **e**, **f** Dynamic force spectroscopy (DFS) plot showing the distribution of the rupture forces as a function of their loading rate (LR) measured either between the S1 subunit and the ACE2 receptor ($N = 1052$ data points) (**e**) or between the RBD and the ACE2 receptor ($N = 1490$ data points) (**f**). The error bar indicates s.d. of the mean value for a single interaction (0–200 pN). The solid line represents the fit of the data with the Bell–Evans fit. Experiments were reproduced at least four times with independent tips and samples. **g**, **h** The BP is plotted as a function of the contact time for S1 subunit and RBD on ACE2 model surfaces, and data points were fitted using a least-squares fit of a monoexponential growth. One data point belongs to the BP from one map acquired at 1 μm/s retraction speed for the different contact times. Experiments were reproduced three times with independent tips and samples. P values were determined by two-sample t test in Origin. The error bar indicates s.d. of the mean value. Source data are provided as a Source Data file.

S1 subunit or RBD- functionalized tip from the ACE2 model surface (Fig. 2a, b). Specific adhesion events were observed on 4–5% of the retraction FD curves at rupture distances >15 nm, which corresponds to the extension of the PEG linker (Fig. 2c and Supplementary Fig. 2), and is in line with studies carried out for other virus–cell-surface receptor systems[8,10–12]. To confirm the specificity of these interactions, we conducted additional independent control experiments using (i) an AFM tip only functionalized with the PEG linker or (ii) toward OH-/COOH-terminated alkanethiol surfaces missing the receptor. The binding frequency observed during those control experiments is significantly lower, thereby confirming the specificity of the S1 subunit/RBD–ACE2 complexes under our experimental conditions (Fig. 2c).

**Exploring the dynamics of S1 subunit–ACE2 interaction.** Single-molecule force-probing techniques, such as FD-based AFM, measure the strength of a bond under an externally applied force, enabling to get insights into the binding free-energy landscape. According to the Bell–Evans model[13,14], an external force stressing a bond reduces the activation-energy barrier toward dissociation and, hence, reduces the lifetime of the ligand-receptor pair[15] (Fig. 2d). The model also predicts that far-from-equilibrium, the binding strength of the ligand-receptor bond is proportional to the logarithm of the loading rate (LR), which describes the force applied on the bond over time. To investigate the kinetics of the probed complex, FD curves were recorded at various retraction rates and contact times (Fig. 2e–h). Dynamic force spectroscopy (DFS) plots were obtained for both S1 subunit (Fig. 2e) and RBD (Fig. 2f) binding toward immobilized ACE2 receptors. In each case, the unbinding force increases linearly with the logarithm of the LR, as observed earlier for other

used FD-curve-based AFM to evaluate at the single-molecule level the binding strength of the interaction established between the glycosylated S1 subunit and ACE2 receptors on model surfaces (Fig. 2a). To mimic cell-surface receptors in vitro, ACE2 receptors were covalently immobilized onto gold surfaces coated with OH- and COOH-terminated alkanethiols using carbodiimide conjugation (see Methods). These model surfaces were imaged by AFM, and the thickness of the grafted layer was validated by a scratching experiment, revealing a deposited layer of 6.1 ± 0.4 nm (mean ± S.D., $N = 3$) (see Methods and Supplementary Fig. 1). To study the interaction between the S1 subunit and the immobilized ACE2 receptors, we covalently grafted either the purified full S1 subunit or RBD only to the free end of a long polyethylene glycol (PEG)$_{24}$ spacer attached to the AFM tip[7–9]. To investigate the properties of the binding complex, force–distance (FD) curves were recorded by repeatedly approaching and withdrawing the

virus-receptor bonds[8,10,11,16,17]. To determine whether single- or multiple-bond rupture between S1/RBD and ACE2 is taking place, bond strengths (every single gray data point in Fig. 2e, f) were analyzed through distinct discrete ranges of LRs, plotted as force histograms and further fitted with multipeak Gaussian distribution, as established previously[11,16] (Supplementary Figs. 3 and 4). Using this distribution, we are able to determine the most probable unbinding force of each force peak (maximum of rupture force distribution; black dots plotted over mean LR of this range in Fig. 2e, f), and can determine if single or multiple interactions were taking place. The presence of multiple parallel unbinding events is first observed in the distribution of rupture forces with the presence of multiple Gaussian fits. The histograms show that most probably only single interactions were taking place; thus, the Bell–Evans model[15] was used to fit the data enabling to interpret the binding complex as a simple two-state model, in which the bound state is separated from the unbound state by a single energy barrier (Fig. 2d). From the slope of the fit, we estimated the length scale of the energy barrier ($x_u$). We obtained very close values, $x_u = 0.81 \pm 0.05$ nm and $0.79 \pm 0.04$ nm for both the S1 subunit and RBD, showing that we are probing similar bonds (Fig. 2e, f). The kinetic off-rate ($k_{off}$) or dissociation rate is obtained from the intercept of the fit (at LR = 0) yielding $k_{off}$ values of $0.008 \pm 0.005$ s$^{-1}$ and $0.009 \pm 0.006$ s$^{-1}$ for S1 subunit and RBD, respectively. These values are in good agreement with reported values obtained by surface plasmon resonance for the S glycoprotein ($k_{off} = 0.003$ s$^{-1}$)[18] and the RBD subunit ($k_{off} = 0.008$ s$^{-1}$) binding to ACE2 receptors[19].

Assuming that the receptor-bond complex can be approximated by a pseudo-first-order kinetics, we also estimated the kinetic on-rate ($k_{on}$) from our single-molecule force spectroscopy experiments[11] (Fig. 2g, h). This association rate is extracted from the binding probability (BP) measured at various contact times, and depends on the effective concentration described as the number of binding partners (ligand + receptor) within an effective volume $V_{eff}$ accessible under free-equilibrium interaction. $V_{eff}$ can be approximated by a half-sphere with a radius including the linker, the viral glycoprotein (S1 subunit or RBD) and the ACE2 receptor. For both the S1 subunit and RBD, we observed that the binding frequency increased exponentially with contact time, and we extracted an interaction time of ~0.250 ms, leading to a $k_{on}$ of $6.4 \times 10^4$ M$^{-1}$ s$^{-1}$ and $8.0 \times 10^4$ M$^{-1}$ s$^{-1}$, respectively. Finally, the dissociation constant $K_D$ is calculated as the ratio between the $k_{off}$ and the $k_{on}$, yielding values around ~120 nM for both complexes. This value corresponds to a high-affinity interaction, confirming the specificity of the complexes established by SARS-CoV-2 with the ACE2 cell-surface receptor, which in turn results in a long lifetime of the virus attachment to the cell surface. Other interaction studies between SARS-CoV (80% sequence homology to SARS-CoV-2) and ACE2 reported specific, high-affinity association values also in the nM range[20]. For comparison, a variety of examples for low- as well as high-affinity interactions between other virus-receptor pairs are summarized in Dimitrov et al.[21] and include influenza A—SA (mM) or HIV-1—CD4 (nM) interactions. For single-molecule interactions, the bond lifetime $\tau$ can be directly related to the inverse kinetic off-rate ($\tau = k_{off}^{-1}$), resulting here in a $\tau$ of 125 ms for the S1 subunit and 111 ms for the RBD, respectively. Of course, at the virion level, the overall bond lifetime will increase with the multivalence of the interaction. By definition, high-affinity interaction has a long lifetime as the dissociation constant $K_D$ is defined as the ratio between $k_{off}$ and $k_{on}$. For high-affinity interactions, the $K_D$ is in the nM range, leading to $k_{off} \ll k_{on}$ and therefore maintaining the interaction in its bond state for very long times, making the development of anti-binding molecules targeting this interaction more difficult. Finally, we also used

optical biolayer interferometry (BLI) to confirm the kinetic parameters characterizing this interaction, and obtained very close affinities in the same nM range as AFM experiments (Supplementary Fig. 5). Taken together, our in vitro experiments confirm that SARS-CoV-2 binding to the ACE2 receptors is mediated by the RBD–ACE2 interface as our experimental conditions did not highlight any significant difference between S1 subunit and RBD binding.

**Validation of the interaction on living cells**. Next, we wanted to investigate whether the interaction probed on isolated receptors is also established in physiologically relevant condition. To this end, we performed binding assays on living A549 cells (human adenocarcinoma alveolar basal epithelial cells). While this cell line is widely used as a type II pulmonary epithelial cell model, it has been shown recently that those cells are incompatible with SARS-CoV-2 infection[22]. Interestingly, ACE2 expression positively correlated with the differentiation state of epithelia. Although undifferentiated cells (cultured at low confluency) only express little ACE2, overexpression of ACE2 in undifferentiated A549 cells facilitated virus entry[23]. We transiently transfected ACE2–eGFP in A549 cells (A549–ACE2) and probed S1-subunit binding to those cells as well as to A549 cells (serving as internal control) (Fig. 3a and Supplementary Fig. 6). Confocal images showed ACE2–eGFP receptors homogeneously distributed in small domains at the surface of A549 cells (Fig. 3b). Guided by fluorescence (Fig. 3c), we chose areas in which both cell types, i.e., transfected (A549–ACE2, green fluorescence) and nontransfected (A549, no fluorescence) cells, were in proximity to one another. Having both A549 cell types in one image area served as a direct control to evaluate whether interactions measured by the functionalized tip were indeed due to specific binding to fluorescent ACE2–eGFP receptors, and to evaluate the extent of other types of interactions (Fig. 3c–e). In such area, we simultaneously recorded a height image (Fig. 3d) and the corresponding adhesion map (Fig. 3e), which were reconstructed from FD curves recorded for each topographic pixel. The retraction part of FD curves showed specific adhesion events mainly on A549–ACE2 cells, with a significantly higher BP (Fig. 3f), as exemplified with the presented adhesion map that shows 20.1% of adhesive pixels on the A549–ACE2 cell versus 13.5% on the control cells (Fig. 3e and Supplementary Fig. 7). Specific binding forces (and corresponding LR) were extracted from force vs. time curves recorded on A549–ACE2 cells (Fig. 3g) and overlaid on the DFS plot obtained on purified ACE2 receptors (Fig. 3h). To explore a wide range of LR, we probed the interaction at various frequencies and amplitudes (see Methods). We observed a very good alignment between the data obtained on purified receptors and on living cells confirming the physiological relevance of our results obtained on model surfaces.

**S1 subunit binding to the cell involves other receptors**. Our FD-based AFM experiments performed on living cells put in evidence that the S1 subunit interacts even on control cells with a frequency ≈10% although the expression level of ACE2 should be very low as the cells are not differentiated. Nevertheless, some evidence pointed out that human CoV S glycoproteins possess sialic acid (SA)-binding sites and in particular to 9-O-acetyl-sialogycans[24], and that integrins could also be a receptor for the SARS-CoV-2 (ref. [25]), which possesses a RGD motif close to the ACE2-binding site. To evaluate whether these other receptors could be involved during the early binding steps to the cell surface, we performed additional experiments by injecting 9-O-acetyl-sialogycans to block interaction with cell-surface SA, or added cyclo-RGD (cRGD) to compete with the interactions with

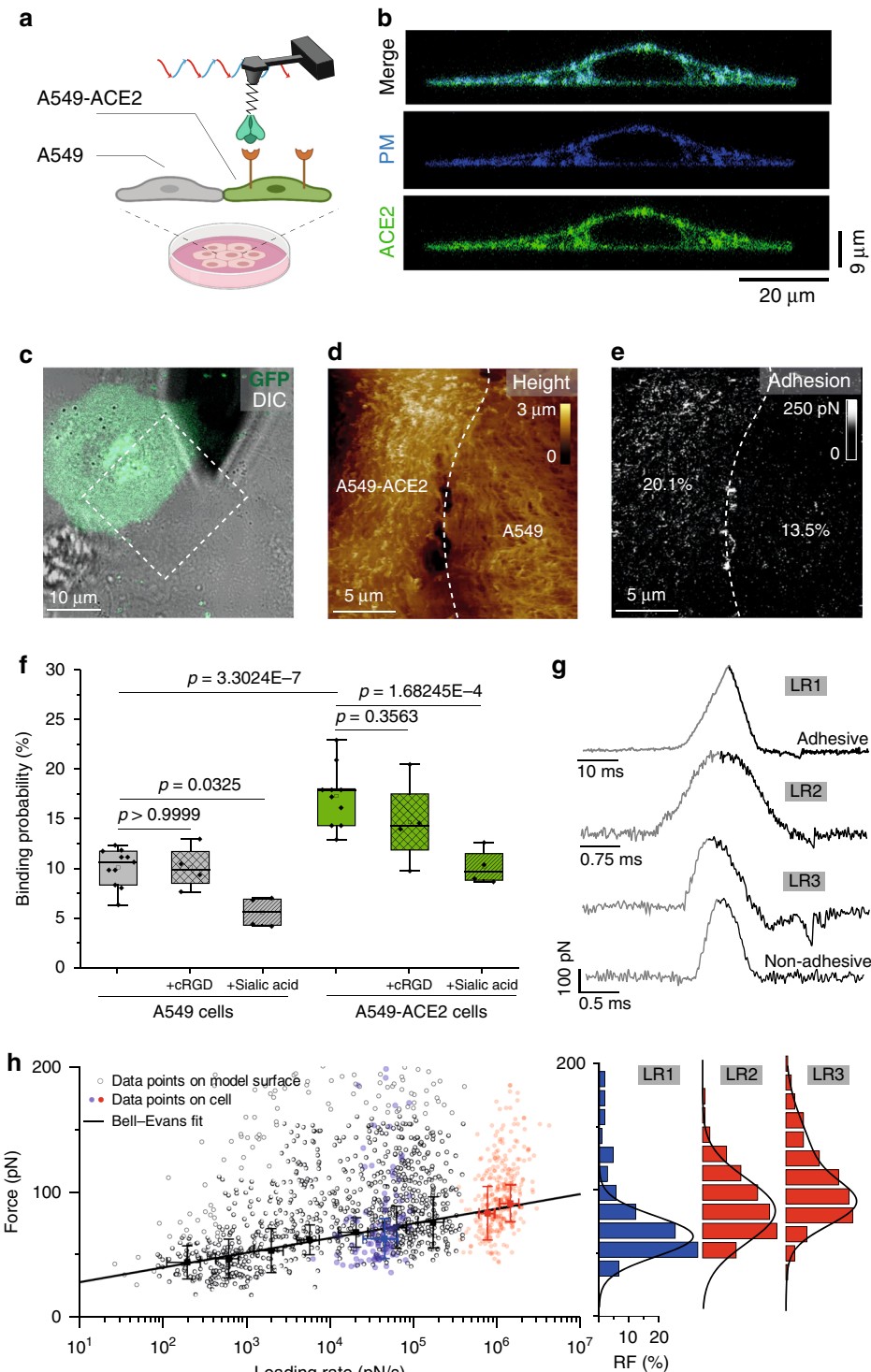

integrins. After SA injection, the binding frequency was reduced on A549 cells down to ~7% and to ~10% on ACE2-transfected cells (Fig. 3f and Supplementary Fig. 8). For integrins, injection of cRGD only reduces the binding frequency of ~1–2% on both cell types, which is in good agreement with the fact that integrins are mostly expressed on the bottom of the cell[26]. Altogether, these data obtained on cells by AFM represent to date the best evidence that S1–ACE2 complex is established in physiologically relevant conditions and underlines the complex situation with multiple cell-surface receptors accounting for the whole interaction.

**Inhibition of S1-subunit binding using ACE2-derived peptides**. Human recombinant soluble ACE2 (hrsACE2) is currently being considered for treatment of COVID-19 (refs. [27,28]). However, ACE2 is involved in many key cellular processes, such as blood-pressure regulation and other cardiovascular functions. Therefore, hrsACE2 treatment could lead to dysregulation of those vital processes and subsequently cause deleterious side effects for treated patients. To avoid any interference of the ACE2 homeostasis, we wanted to test whether small ACE2-derived peptides can also interfere with SARS-CoV-2 binding, by blocking binding

**Fig. 3 Probing S-glycoprotein binding to the ACE2 host receptor on living cells. a** Binding of S-glycoprotein subunit 1 (S1) is probed on A549 and A549–ACE2 cells. **b** Confocal microscopy (z stack) of A549–ACE2–eGFP (green) cell transduced with plasma membrane BFP (blue). **c** Overlay of eGFP and DIC images of a mixed culture of A549 and A549–ACE2–eGFP cells. **d**, **e** Force–distance (FD)-based AFM topography image (**d**) and the corresponding adhesion map (**e**) in the specified area in (**c**). The frequency of adhesion events is indicated. **f** Box plot of the binding probability between S1 and A549 cells (gray) or A549–ACE2 cells (green) without and after injection of cyclic RGD (cRGD, checked boxes) or sialic acid (SA, dashed boxes), respectively. The square in the box indicates mean, the colored box indicates the 25th and 75th percentiles, and the whiskers indicate the highest and the lowest values of the results. The line in the box indicates median. **g** Force versus time curves showing either a nonadhesive curve (bottom) or specific adhesive curves acquired at different LRs (LR1–LR3). **h** DFS plot showing the distribution or the rupture forces measured either between the S1 subunit and the ACE2 on model surfaces (black dots, extracted from Fig. 2e), and between the S1 subunit and ACE2-overexpressing A549 cells acquired at three different LRs (blue and red dots) ($N = 403$). Blue dots belong to a data set acquired in fast-force volume mode, with a retraction velocity of $20\,\mu m\,s^{-1}$ (LR1). Red dots belong to data sets acquired in peak force tapping mode with 0.125 kHz peak force frequency and 375-nm amplitude (LR2) or at 0.25 kHz and 750 nm (LR3), respectively. The error bar indicates s.d. of the mean value. Histograms of force distribution on A549–ACE2 cells for LR1–LR3 are shown on the side. For experiments without injection of cRGD or SA, data are representative of at least $N = 11$ cells from $N = 6$ independent experiments. The data for blocking experiments with cRGD or SA were acquired for at least $N = 4$ cells from $N = 2$ independent experiments. $P$ values were determined by two-sample $t$ test in Origin. Source data are provided as a Source Data file.

sites on the S glycoprotein. To this end, we synthetized four different peptides (sequences provided in Supplementary Fig. 9), which have been selected to mimic the regions of ACE2 that interact with the S1 subunit as determined by the crystal structure[29], and we tested their binding inhibition properties using our single-molecule force spectroscopy approach (Fig. 4a, b). We first measured the BP between the S1 subunit and the ACE2 in the absence of peptide (0 μM), with a contact time of 250 ms, as reference, and then injected our ACE-derived peptides at three different concentrations (1, 10, and 100 μM). For the four peptides, we observed a progressive reduction of the BP as a function of the concentration confirming a specific inhibition. In addition, for each peptide, we noticed a reduction of >50% of the probed interactions already for the 1–10 μM concentration, suggesting a 50% inhibitory concentration ($IC_{50}$) in the μM range. The [22–44] peptide shows the highest inhibition of the S1–ACE2 complex formation with a measured reduction in the BP of ~76%. The [22–57] peptide shows a similar inhibition potential (~74%), suggesting that the additional amino acids do not influence the overall affinity of the peptide for the S1 subunit, as also confirmed by molecular dynamics (MD) simulations showing that although the peptide 22–57 is longer, less H bonds are established between the peptide and the RBD domain (Supplementary Fig. 10). Overall, these results are in good agreement with the structural insights because these peptides are derived from the N-terminal helix of the ACE2 and therefore form with the RBD interface an important network of hydrophilic interactions (including nine hydrogen bonds and a salt bridge). Within the ACE2–RBD complex, the [351−357] fragment is also part of a "hot binding spot" that results in our test by a good score with a reduction of ~60% of the initial specific BP. Finally, the [22–44–g–351–357] peptide was also synthetized and tested based on the fact that in the crystal structure, the distance between S44 and L351 is close enough to be filled by a single amino acid. A glycine residue was added between the two fragments because the two ACE2 fragments have opposite directionality, and glycine has a high propensity to form reverse turns. Nevertheless, under our experimental conditions, we did not notice any strong improvement in the binding inhibition. Altogether, our in vitro assays at the single-molecule level provide direct evidence that ACE2-derived peptides are strong candidates to potentially inhibit SARS-CoV-2 binding to ACE2 receptors (Fig. 4c).

**ACE2-derived peptide blocks specific binding to living cells.** Finally, we tested whether the [22−57]-binding inhibition peptide could also prevent S1-subunit binding in the cellular context (Fig. 5). The interaction between the S1 subunit and the confluent layer of a coculture of A549 and A549–ACE2 cells was probed

before and after addition of the peptide at 100 μM. Before injection, cells overexpressing the ACE2 receptors (A549–ACE2) show higher BP ($9.4 \pm 1.6\%$ vs. $19.4 \pm 7.3\%$, for A549 and A549–ACE2, respectively) (mean ± S.D., $N = 4$) (Fig. 5a–d), in good agreement with our previous observation (Fig. 3f). After injection of the [22–57] ACE2-derived peptide, we observed a significant decrease of the BP on both cell types (Fig. 5e, f). In particular, the BP on A549–ACE2 cells significantly drops (~70%), reaching a level close to the one of the control cells. Taking into account that undifferentiated A549 cells express little ACE2 and are poorly infected by CoV[23], this result supports the biological relevance of our ACE2-derived peptide acting as potential inhibitor capable of efficiently blocking SARS-CoV-2 binding.

In conclusion, we investigated the interaction established between the SARS-CoV-2 S glycoprotein and the ACE2 receptor using single-molecule force spectroscopy. We demonstrated a specific binding mechanism between the S1 subunit and the ACE2 receptor. By comparing the binding of the S1 subunit and the RBD toward the ACE2 receptor, our experiment evidenced that both domains interact with the same kinetic and thermodynamic properties toward the ACE2 receptor, highlighting that SARS-CoV-2 binding to ACE2 is dominated by the RBD/ACE2 interface. Our measurements show that under our physiologically relevant conditions, the RBD binds the ACE2 receptor with an intrinsic high affinity (~120 nM), which could even be further stabilized at the whole-virus level, thanks to possible multivalent bonds between the S-glycoprotein trimer and ACE2 dimer.

Based on the available crystal structures of the molecular complex, we examined how several ACE2-derived peptide fragments could interfere with the S1–ACE2 complex formation. While all tested peptides show binding inhibition properties, peptides mimicking the N-terminal helix of the ACE2 receptor show the best results. Both [22–44] and [22–57] peptides exhibit an anti-binding activity with $IC_{50}$ in the μM range, resulting in a >70% decrease in the BP observed by AFM on purified receptor and >70% on living cells. On the cellular model, we observed that the BP drops to the level of the control cells (undifferentiated A549 cells) that are poorly infected by CoV[23]. Therefore, those peptides appear as strong therapeutic candidates against the SARS-CoV-2 infection.

## Methods

**Cell culture and transfection**. A549 cells (ATCC® CCL-185) were grown in Ham's F-12 Nutrient Mix with 10% fetal bovine serum, penicillin ($100\,U\,ml^{-1}$), and streptomycin ($100\,\mu g\,mL^{-1}$) (Gibco) at 37 °C in a humidified atmosphere with 5% $CO_2$. pcDNA3.1(+) ACE2–eGFP was transfected using Lipofectamine LTX (Invitrogen) according to the manufacturer's protocol. In brief, 2 μg of

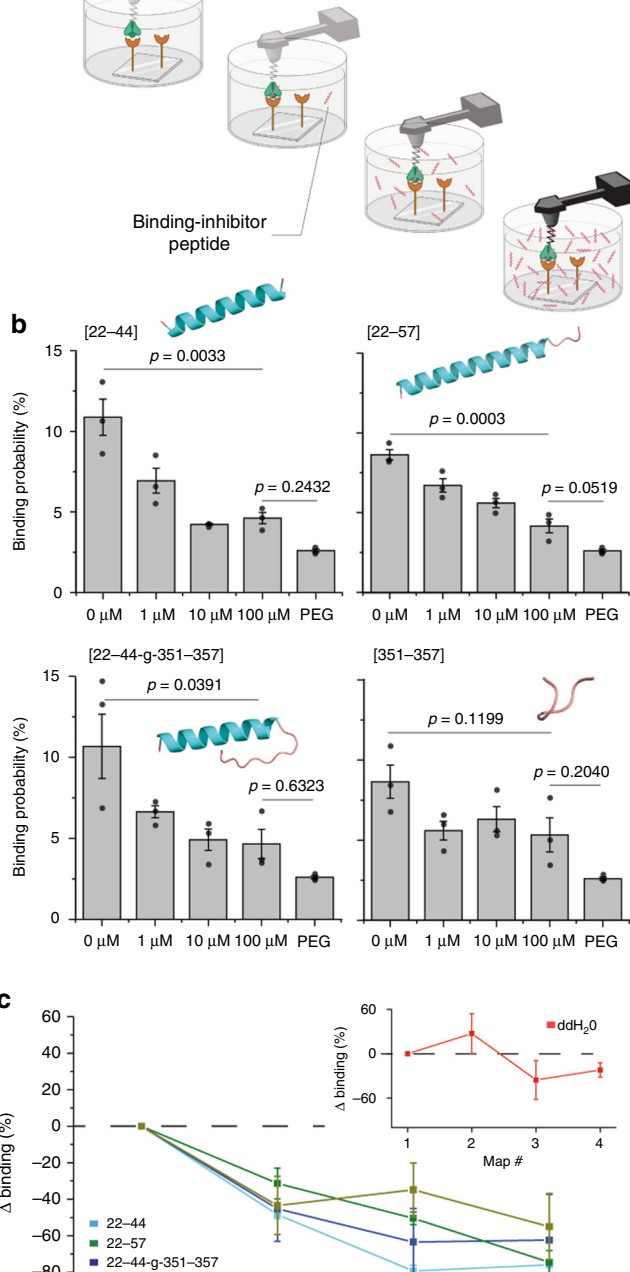

**Fig. 4 Anti-binding effects of ACE2-derived peptides on S1-subunit binding. a** Efficiency of blocking peptides is evaluated by measuring the binding probability of the interaction between the S1 subunit and ACE2 receptor on model surface before and after incubation of the functionalized AFM tip with the four different peptides at increasing concentration (1–100 μM). **b** Histograms, with the corresponding data points overlaid in dark gray, showing the binding probability without peptide (0 μM) and upon incubation with 1, 10, or 100 μM of ACE2-derived peptides ([22–44], [22–57], [22–44–g–351–357], and [351–357]). The binding probability measured with a polyethylene glycol (PEG) tip enables to evaluate the nonspecific binding level. The prediction of the structure of the ACE2-derived peptides is shown in the inset. The structure of the peptides is based on the structure of the peptide in the crystal structure (PDB ID: 6m0j). For the [22–44–g–351–357] peptide, its structure was generated using homology modeling[41]. The error bar indicates s.d. of the mean value. **c** Graph showing the reduction of the binding probability. Control with ddH$_2$O is provided in the inset showing that repetitive measurements do not result in a similar decrease of the binding probability. Data are representative of at least $N = 3$ independent experiments (tips and sample) per peptide concentration. *P* value was determined by two-sample *t* test in Origin. The error bar indicates s.d. of the mean value. Source data are provided as a Source Data file.

pcDNA3.1(+) ACE2–eGFP was transfected to A549 cells (60-mm plate) using 6 μl of Lipofectamine LTX and 2 μl of PLUS reagent (Invitrogen).

**Functionalization of AFM tips.** PFQNM-LC and MSCT-D cantilevers (Bruker) were used to probe the interaction between S1 subunit (Genscript, #U5377FC120) or RBD protein (Genscript, #U5377FC120) and ACE2 protein (Sino Biological, 90211-C02H). NHS-PEG$_{24}$-Ph-aldehyde linkers were used to functionalize AFM tips as previously described[30]. Briefly, the cantilevers were immersed in chloroform for 10 min and further cleaned in a UV radiation and ozone (UV-O) cleaner (Jetlight), and immersed overnight in an ethanolamine solution (3.3 g of ethanolamine in 6.6 ml of DMSO). They were washed with DMSO and ethanol three times, respectively. Ethanolamine-coated cantilevers were immersed in NHS-PEG$_{24}$-Ph-aldehyde solution (3.3 mg of it was diluted in 0.5 ml of chloroform and

30 μl of triethylamine) and finally washed 3 times with chloroform and dried with nitrogen.

For AFM tips functionalized with S1-subunit protein, 50 μl of S1-subunit protein solution (0.1 mg/ml) was put onto the cantilevers placed on Parafilm (Bemis NA) and 2 μl of fresh NaCNBH$_3$ solution (6 wt% vol-1 in 0.1 M NaOH(aq)) was mixed in the protein solution. The cantilevers were incubated in the solution for 1 h on ice. Then, 5 μl of 1 M ethanolamine solution was carefully added to the protein solution and incubated 10 min to quench the reaction and finally washed three times with PBS.

For AFM tips derivatized with the RBD protein, 100 μl of a 100 μM tris-nitrilotriacetic amine 540 trifluoroacetate (Toronto Research Chemicals, Canada) (tris-NTA) solution was put onto them placed on Parafilm, and 2 μl of fresh NaCNBH$_3$ solution was mixed in the protein solution. They were incubated in the solution for 1 h on ice. Then, 5 μl of 1 M ethanolamine solution in the protein solution was added and incubated for 10 min. The mixture of 50 μl of RBD solution (0.1 mg ml$^{-1}$) and 2.5 μl of 5 mM NiCl$_2$ were put onto them and they were incubated for 2 h. After incubation, they were washed in PBS solution three times.

**Preparation of ACE2-coated model surfaces.** ACE2 protein (Sino Biological, 90211-C02H) was immobilized using NHS–EDC chemistry. Gold-coated surfaces were first rinsed with ethanol, dried with a gentle stream of nitrogen gas, cleaned for 15 min by UV and ozone treatment (Jetlight), and incubated overnight in an alkanethiol solution (99% 11-mercapto-1-undecanol 1 mM (Sigma Aldrich) and 1% 16-mercaptohexadecanoic acid 1 mM (Sigma Aldrich) in ethanol). The chemically activated samples were rinsed with ethanol, dried with nitrogen gas, and immersed for 30 min in the solution of 100 mg of chemically activated dimethyl-laminopropyl carbodiimide (Sigma Aldrich) and 40 mg of N-hydroxysuccinimide in 4 ml of milliQ water. Finally, the surfaces were rinsed with milliQ water, incubated with ACE2 protein (0.1 μg μL$^{-1}$ in PBS) on Parafilm (Bemis NA), and washed in PBS.

**FD-based AFM on model surfaces.** FD-based AFM on model surfaces was performed in PBS at room temperature using functionalized MSCT-D probes (Bruker, nominal spring constant of 0.030 N/m and actual spring constants calculated using thermal tune)[31]. A Bioscope Resolve AFM (Bruker) operated in the force–volume (contact) mode (Nanoscope software v9.1) was used. Areas of 5 × 5 μm were scanned, ramp size set to 500 nm, and set point force of 500 pN, with a resolution of 32 × 32 pixels and a line frequency of 1 Hz.

DFS analysis (using a constant approach speed of 1 μm/s and variable retraction speeds of 0.1, 0.2, 1, 5, 10, and 20 μm/s) and kinetic on-rate estimation (measuring the BP for different hold times of 0, 50, 100, 150, 250, 500, and 1000 ms) were performed. Regarding DFS experiments, data including LRs and disruption forces were extracted using Nanoscope analysis (v2.0, Bruker). Origin software (OriginLab) was used to display the results in DFS plots to fit histograms of rupture force distributions for distinct LR ranges, and to apply various force spectroscopy models, as described[8,16]. For kinetic on-rate analysis, the BP (fraction of curves showing binding events) was determined at a certain hold time (t) (the time the tip is in contact with the surface). Those data were fitted and $K_D$ calculated as described previously[32]. In brief, the relationship between interaction time ($\tau$) and

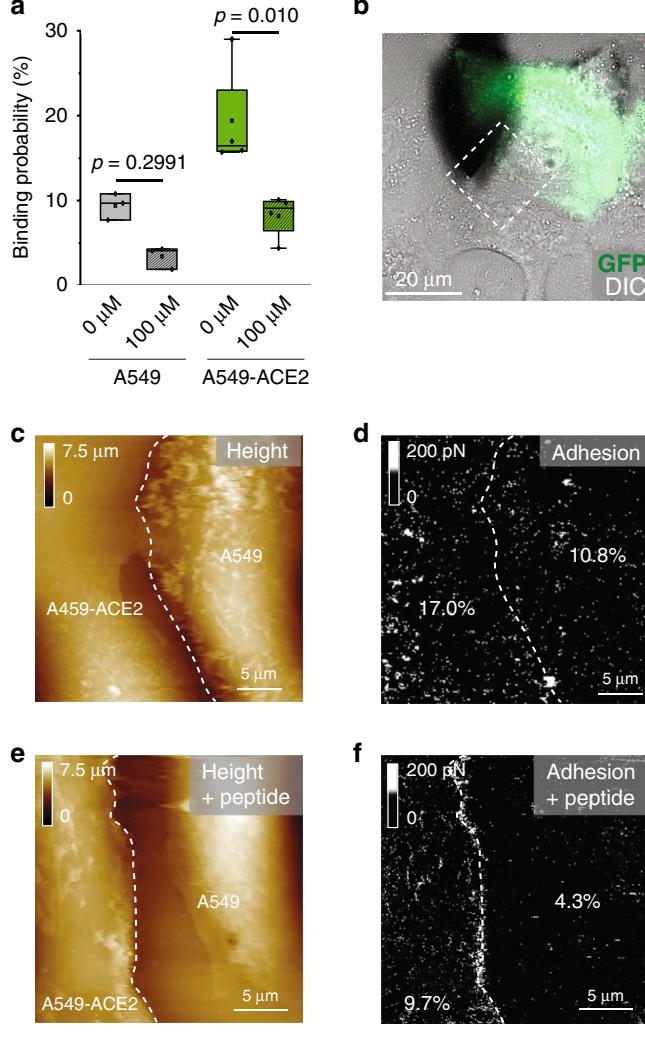

**Fig. 5 Blocking of S1-subunit binding using ACE2-derived peptide on living cells. a** Box plot showing that the reduction of binding probability measured the S1-subunit-derivatized tip and a mixed culture of A549 and A549–ACE2 cells upon injection of the [22–57] ACE2-derived peptide. The square in the box indicates mean, the colored box indicates the 25th and 75th percentiles, and the whiskers indicate the highest and the lowest values of the results. **b** Overlay of eGFP and DIC images of a mixed culture of A549 and A549–ACE2–eGFP cells. FD-based AFM topography images (**c**, **e**) and the corresponding adhesion map (**d**, **f**) recorded in the specified area in (**b**) (scanned with a scan angle) before (**c**, **d**) and after (**e**, **f**) incubation of the tip with the [22–57] ACE2-derived peptide. The frequency of adhesion events is indicated. Data are representative of at least $N = 4$ cells from $N = 2$ independent experiments. $P$ values were determined by two-sample $t$ test in Origin. Source data are provided as a Source Data file.

BP is described by the following equation:

$$BP = A * \left[ 1 - \exp\left( \frac{-(t - t_0)}{\tau} \right) \right] \quad (1)$$

where $A$ is the maximum BP and $t_0$ the lag time. Origin software is used to fit the data and extract $\tau$. In the next step, $k_{on}$ was calculated by the following equation, with $r_{eff}$ the radius of the sphere, $n_b$ the number of binding partners, and $N_A$ the Avogadro constant

$$k_{on} = \frac{\frac{1}{2} \cdot 4\pi r_{eff}^3 N_A}{3 n_b \tau} \quad (2)$$

The effective volume $V_{eff}$ ($4\pi r_{eff}^3$) represents the volume in which the interaction can take place. This results in a half-sphere, since only half of the S1 molecules can interact with its corresponding receptor on the substrate.

**Peptides and competition-binding assays.** To assess the influence of peptides on the S1-subunit–ACE2 interaction, binding probabilities were measured before and after tip incubation with 1, 10, and 100 μM of peptide. Briefly, a first map was recorded as described above (i.e., force-volume mode, 1 μm/s approach and retraction speed, ramp size of 500 nm, an applied force of 500 pN, resolution of $32 \times 32$ pixels, line frequency of 1 Hz, and hold time of 250 ms), then the peptide at the appropriate concentration was injected, and a new map was recorded.

All the peptides ([22–44], [351–357], [22–57], and [22–44–g–351–357]) were synthesized by Genscript (Hong Kong). Those peptides are designed according to the sequence of the ACE2 receptor in complex with the RBD domain of the S1 glycoprotein.

**FD-based AFM and fluorescence microscopy on living cells.** An AFM (Bioscope Resolve, Bruker) coupled to a confocal microscope (Zeiss LSM-980) was used to acquire correlative images. The AFM was equipped with a 150-μm piezoelectric scanner. The AFM and the microscope were equipped with a cell-culture chamber allowing maintaining the temperature ($37 \pm 1$ °C). To keep cells alive, the humidified ($95 \pm 2\%$ relative humidity) synthetic air (80% $N_2$ and 20% $O_2$) was supplemented with 5% $CO_2$ and filled continuously around the cell plate allowing to diffuse into cell-culture media[16]. Fluorescence images were recorded using a water-immersion lens (×63, NA 1.20, Zeiss C-Apochromat). PFQNM-LC cantilevers (Bruker) were used to record AFM images (~25 μm²) at imaging forces of ~500 pN. The cantilevers were oscillated either at 0.25-kHz peak force frequency with a 750-nm amplitude, 0.125 kHz with a 375-nm amplitude in the PeakForce Tapping mode, or at 20 μm s⁻¹ retraction speed in fast-force-volume mode. The sample was scanned using 256 pixels per line (256 lines) and a frequency of 0.125 Hz. To study the involvement of other receptors, cells were treated with either 1 mM of 9-O-acetyl-sialogycans or 500 μM of cRGD. AFM images and FD curves were analyzed using Nanoscope analysis software, Origin, Gwyddion, and ImageJ. Optical images were analyzed using Zen software (Zeiss). The fluorescence intensity was measured with Zen software (Zeiss). The same size of the area was taken on A549–ACE2 and A549 cells. The average intensity of the area was calculated with Zen software. The statistical analysis was performed with Prism (Graphpad).

**Plasma membrane staining.** Plasma membrane-CFP BacMam 2.0 (Invitrogen) was used to check the co-localization of ACE2 protein and plasma membrane according to the manufacturer's protocol. In brief, 2 μl of plasma membrane-CFP BacMam 2.0 per 10,000 cells was added on the cell-culture dish 16 h (37 °C) before imaging. Z-stack image was recorded by confocal LSM-980 (Zeiss) using a water-immersion lens (×63, NA 1.20, Zeiss C-Apochromat) and 445- and 488-nm laser line.

**Affinity measurements using BLI.** Affinity between the S1 subunit or RBD and ACE2 was also investigated by BLI, using a Blitz® device equipped with amine-reactive second-generation (AR2G) biosensors (Pall ForteBio). After hydrating the biosensor for 10 min and performing an initial baseline (1 min), the biosensor surface was chemically activated (5 min) by a freshly prepared 20 mM EDC and 10 mM NHS (in milliQ water) solution. Then, ACE2 (0.025 μg μL⁻¹ in acetate buffer, pH 4) was loaded onto the biosensor during 3 min and the reaction quenched with ethanolamine 1 M (pH 8). After another baseline step (1 min in PBS), binding of S1 subunit or RBD (0.1 mg mL⁻¹) was measured for 5 min. Finally, the dissociation step (5 min) was performed in PBS. Data processing and analysis were run using a routine provided by GraphPad Prism.

**MD simulation between ACE2 peptides and S glycoprotein.** The PDB (code: 6m0j)[29] was used to perform a MD simulation between ACE2-derived peptides and the SARS-CoV-2 spike protein complex. MD simulations were performed utilizing the Gromacs package[33,34] and carried out using the Amber99SB-ILDN[35] force fields in TIP3P water[36]. The simulation system consisted of a peptide, a protein, and water (about 20,000 molecules) in a cubic box that extended 10 nm from the protein. Appropriate amounts of sodium/chlorine ions were added in the system. For starting the simulation, the environment had to be developed as follows. The steepest descent algorithm was performed either up to 50,000 steps or by 100 kJ mol⁻¹ nm⁻¹. Then, the environment of the system changed at 300 K (NVT ensemble) and subsequently at 300 K and 1 bar (NPT ensemble). After developing the environment, the Particle Mesh Ewald[37] method was used to calculate the long-range electrostatic interactions. Short-range dispersion interactions were described by a Lennard–Jones potential with the cutoff of 1 nm. After reaching the equilibrium of temperature and pressure, MDs were conducted for 60 ns at 300 K and 1 bar. The LINCS algorithm[38] was applied to constrain the covalent bonds with hydrogen atoms. The time step of the simulations was set to 2 fs. The interactions above 10 Å were regarded as nonbond. To determine whether a hydrogen bond exists between a peptide and a protein in the MD models, a geometrical criterion was adopted, in which the formation of a hydrogen bond was defined by both atom distance and bond orientation. For example, assuming donor D, hydrogen H, and acceptor A consists of D–H ··· A configuration. Then when the distance between donor D and acceptor A was shorter than 3.5 Å as well as the bond angle H–D ··· A smaller than 60.0°, it has been regarded as a hydrogen bond. The hydrogen bonds are counted for 55–60 ns while running the simulations.

**Synthesis of 9-O-acetyl-2-α-O-propargyl-SC.** 2-α-O-propargyl SC was synthesized by the protocol described by Dashkan et al.[39]. This molecule was selectively acetylated at the 9-position following the procedure of Ogura et al.[40].

**Reporting summary**. Further information on research design is available in the Nature Research Reporting Summary linked to this article.

## Data availability
The Source data underlying Figs. 2c, e–h, 3f, h, 4b, c, 5a and Supplementary Figs. 2, 6, 7 are provided as a Source Data file. All other relevant data are available from the corresponding authors upon reasonable request. Source data are provided with this paper.

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

## Acknowledgements
This work was supported by the Université catholique de Louvain, the Foundation Louvain, and the Fonds National de la Recherche Scientifique (FRS-FNRS). This project received funding from the European Research Council under the European Union's Horizon 2020 research and innovation program (grant agreement No. 758224) and from the FNRS-Welbio (Grant # CR-2019S-01). The funders had no role in study design, data collection and analysis, decision to publish, or preparation of the paper. S.P., A.C.D., and D.A. are research fellow, postdoctoral researcher, and research associate at the FNRS, respectively. Q.Z., W.C., and S.P.V. are grateful to China Scholarship Council. Cartoons in Figures 1a–c, 2a, d, 3a, and 4a were created with BioRender.com.

## Author contributions
J.Y., S.J.L.P., M.K., A.C.D., and D.A. conceived the project, planned the experiments, and analyzed the data. J.Y., S.J.L.P., and S.D. conducted the AFM experiments. Q.Z. performed MD simulation and structure predictions. S.P., M.K., and P.S. conducted and analyzed the BLItz experiments. W.C. and S.P.V. conceived and synthesized the SA derivative. All authors wrote the paper.

## Competing interests
The authors declare no competing interests.
