## [Peer Review File · Nature Communications]

Reviewers' Comments:

Reviewer #1:

Remarks to the Author:

The authors answered satisfactory for most of my questions, but I have just a few remaining doubts. In question 1, they say that in this single molecule approach the goal is to maintain the binding probability low in order to favor single bond rupture for extracting the physical parameters. Do the authors keep the binding probability low in purpose or it just happens?

In their response to question 2, they give a variety of reasons to justify single molecule techniques against ensemble studies. However, after all these justifications, the single molecule experiments reveal the kinetic parameters that are already known by plasmon resonance. In fact, they present this agreement as a positive support to their experiments. What would happen if the results are different numbers? In this particular case, can the authors add any other new information coming from the fact of using their single molecule technique?

I think this a nice work that deserves to be published in NC, but I still wonder if it could add something else to understand the high infectivity of SARS-CoV-2 compared with other coronaviruses.

Reviewer #2:

Remarks to the Author:

All my previous questions and comments were answered and explained by the authors. The quality of the manuscript was also improved by following the comments and suggestions of all reviewers. I recommend publication.

Point-by-Point Response to the Reviewers Comments

Reviewer #1 (Remarks to the Author):

The authors answered satisfactory for most of my questions, but I have just a few remaining doubts.

Authors: Thank you for your encouraging and constructive comments. Below we have explained point-by-point how we have addressed these comments in our second revision.

1) In question 1, they say that in this single molecule approach the goal is to maintain the binding probability low in order to favor single bond rupture for extracting the physical parameters. Do the authors keep the binding probability low in purpose or it just happens?

Authors: Our tip- and sample functionalization protocols are especially designed to probe single interactions, i.e. max. 2-3 molecules attached to the tip and the molecules coupled to the surface forming a monolayer with almost no aggregates.

2) The authors nicely obtain the kinetic parameters of the binding, such as k_{on} and k_{off} , and compare with the reported values obtained by surface plasmon resonance (line 111). What is the novelty of single molecule biophysics beyond this comparison? In line 124 the authors claim that the dissociation constant KD corresponds to high affinity interaction. Compared with what? They affirm that "results in a long lifetime of the virus attachment to the cell surface". Which is the value of this lifetime? Please, provide more insights here.

Authors: The reviewer is indeed right, and our single-molecule experiments reveal kinetic parameters that are the same order of magnitude as the ones measured by other techniques, such as surface plasmon resonance. Single-molecule approaches are nowadays revolutionizing modern biosciences due to the unprecedented insights into complex biological systems. According to the ergodicity hypothesis from statistical mechanics, a sufficiently long time average (or sufficient number of observations) from a single molecule is equivalent to a standard population-averaged snapshot, suggesting that, in principle, a single-molecule measurement contains all information of the molecular ensemble. AFM-based single-molecule approaches work at the low numbers found for most specific proteins in a living cell (typically 1-1000), which eliminates the need for artificial enrichment, such as for ensemble techniques. Moreover, single-molecule measurements enable the quantitative measurement of the kinetics of complex processes without the need for a perturbing synchronization of molecules to reach a sufficient ensemble-averaged signal. We have also

shown how single-molecule experiments allow a precise localization (with nanometer accuracy) and counting of molecules in spatially distributed samples, such as living cells, which is not possible in surface plasmon resonance or typical bulk techniques. Thus, the novelty of this approach when compared to bulk biophysical techniques is that our AFM-based single-molecule method allows us to directly localize and quantify kinetic and thermodynamic parameters of virus-cell receptor interactions in physiological conditions on the surface of living cells. When working at the single-molecule level, sample heterogeneities or rare-transient species along a reaction pathway can be revealed, which are usually averaged out in ensemble measurements. These unique states or events could indeed give rise to different kinetic parameters than the ones characteristic to a larger population of molecules.

Regarding the second part of his question, we compare the high affinity interaction with other interaction studies between SARS-CoV and ACE2, as well as other virus-receptor pairs. In addition, we provide the calculation of the bond life time as well as more insights on this result in lines 139-153: "This value corresponds to a high affinity interaction, confirming the specificity of the complexes established by SARS-CoV-2 with the ACE2 cell surface receptor, which in turn results in a long lifetime of the virus attachment to the cell surface. Other interaction studies between SARS-CoV (80% sequence homology to SARS-CoV-2) and ACE2 reported specific, high-affinity association values also in the nM range¹. For comparison, a variety of examples for low as well as high-affinity interactions between other virus-receptor pairs are summarized in Dimitrov et al.² and include influenza A – sialic acid (mM) or HIV-1 – CD4 (nM) interactions. For single molecule interactions the bond lifetime τ can be directly related to the inverse kinetic off rate ($\tau = k_{\text{off}}^{-1}$) resulting here in a τ of 125 ms for the S1-subunit and 111 ms for the RBD, respectively. Of course, at the virions level, the overall bond lifetime will increase with the multivalence of the interaction. By definition, high-affinity interaction has a long lifetime as the dissociation constant K_D is defined as the ratio between k_{off} and k_{on} . For high-affinity interactions, the K_D is in the nM range leading to $k_{\text{off}} \lllll k_{\text{on}}$ and therefore maintaining the interaction in its bond state for very long times, making the development of anti-binding molecules targeting this interaction more difficult."

1 Li, W. *et al.* Angiotensin-converting enzyme 2 is a functional receptor for the SARS coronavirus. *Nature* **426**, 450-454 (2003).

2 Dimitrov, D. S. Virus entry: molecular mechanisms and biomedical applications. *Nat. Rev. Microbiol.* **2**, 109-122 (2004).

3) I think this a nice work that deserves to be published in NC, but I still wonder if it could add something else to understand the high infectivity of SARS-CoV-2 compared with other coronaviruses.

Authors: We thank the reviewer for this interesting question, if this approach allows to understand the high infectivity of SARS-CoV-2 compared to other coronaviruses. However,

as already explained in the previous revision this study is beyond the scope of this publication. In the study presented here, our aim is to maintain the binding probability low in order to favor single bond rupture which allows the extraction of the kinetics and thermodynamics. Therefore, using this experimental approach we are not able to directly probe the degree of infectivity on a quantitative manner as other steps influence the infection. However, virus specific binding to cell surface is a prerequisite for efficient infection, and therefore of crucial importance, as it helps to concentrate the virus on the cell surface and to promote consecutive steps towards cell infection.